# Surface-Enhanced Raman Analysis of Uric Acid and Hypoxanthine Analysis in Fractionated Bodily Fluids

**DOI:** 10.3390/nano13071216

**Published:** 2023-03-29

**Authors:** Furong Tian, Luis Felipe das Chagas e Silva de Carvalho, Alan Casey, Marcelo Saito Nogueira, Hugh J. Byrne

**Affiliations:** 1FOCAS Research Institute, Technological University Dublin Camden Row, D08CKP1 Dublin, Ireland; alan.casey@tudublin.ie (A.C.);; 2Centro Universitario Braz Cubas, Mogi das Cruzes 08773-380, Brazil; 3Universidade de Taubate, Taubate 12080-000, Brazil; 4Tyndall National Institute, Lee Maltings Complex, Dyke Parade, T12R5CP Cork, Ireland; marcelo.nogueira@tyndall.ie; 5Department of Physics, University College Cork, College Road, T12K8AF Cork, Ireland

**Keywords:** vibrational spectroscopy, bodily fluids, blood serum, saliva, urine, surface-enhanced Raman spectroscopy, gold nanostars, centrifugal filtration, uric acid, hypoxanthine

## Abstract

In recent years, the disease burden of hyperuricemia has been increasing, especially in high-income countries and the economically developing world with a Western lifestyle. Abnormal levels of uric acid and hypoxanthine are associated with many diseases, and therefore, to demonstrate improved methods of uric acid and hypoxanthine detection, three different bodily fluids were analysed using surface-enhanced Raman spectroscopy (SERS) and high-performance liquid chromatography (HPLC). Gold nanostar suspensions were mixed with series dilutions of uric acid and hypoxanthine, 3 kDa centrifugally filtered human blood serum, urine and saliva. The results show that gold nanostars enable the quantitative detection of the concentration of uric acid and hypoxanthine in the range 5–50 μg/mL and 50–250 ng/mL, respectively. The peak areas of HPLC and maximum peak intensity of SERS have strongly correlated, notably with the peaks of uric acid and hypoxanthine at 1000 and 640 cm^−1^, respectively. The r^2^ is 0.975 and 0.959 for uric acid and hypoxanthine, respectively. Each of the three body fluids has a number of spectral features in common with uric acid and hypoxanthine. The large overlap of the spectral bands of the SERS of uric acid against three body fluids at spectra peaks were at 442, 712, 802, 1000, 1086, 1206, 1343, 1436 and 1560 cm^−1^. The features at 560, 640, 803, 1206, 1290 and 1620 cm^−1^ from hypoxanthine were common to serum, saliva and urine. There is no statistical difference between HPLC and SERS for determination of the concentration of uric acid and hypoxanthine (*p* > 0.05). For clinical applications, 3 kDa centrifugal filtration followed by SERS can be used for uric acid and hypoxanthine screening is, which can be used to reveal the subtle abnormalities enhancing the great potential of vibrational spectroscopy as an analytical tool. Our work supports the hypnosis that it is possible to obtain the specific concentration of uric acid and hypoxanthine by comparing the SER signals of serum, saliva and urine. In the future, the analysis of other biofluids can be employed to detect biomarkers for the diagnosis of systemic pathologies.

## 1. Introduction

In purine catabolism, xanthine oxidase oxidises hypoxanthine to xanthine and then xanthine to uric acid. The balance between the uric acid synthesis and degradation maintains uric acid levels in blood serum. Abnormal levels of uric acid and hypoxanthine in serum and urine are associated with defects in enzymes in purine metabolism [1] and are reported in cardiovascular diseases, gouty arthritis, renal disease, type 2 diabetes, obesity and hypertension [2]. In most research studies, colorimetric methods are used to determine urinary creatinine levels [2,3]. A more complex method using high performance liquid chromatography (HPLC) has been used to analyse uric acid and hypoxanthine [4]. Operation of liquid chromatography (LC) and mass spectrometry (MS/MS) are costly and time consuming [5]. The potential of vibrational spectroscopic techniques for biomedical applications has been well demonstrated [6]. Both infrared absorption and Raman scattering spectroscopies provide a label free molecular fingerprint of the sample and have been extensively explored for non-destructive/non-invasive and real-time biofluid analysis [6]. In the analysis of blood serum, Raman spectroscopy has been shown to have practical advantages over infrared absorption spectroscopy, and metal nanoparticles can induce further surface-enhanced Raman scattering (SERS) for such applications [7]. The advantages include Raman scattering spectroscopies not being strongly affected by water signals in the fingerprint region, and the variety of molecular features identified by sharp peaks in post-processed Raman scattering spectroscopies measurements (after background subtraction and possibly normalisation, in studies comparing results of multiple experiments) [8].

SERS analysis of biofluids holds great potential for diagnostics, as suggested by the high accuracies, sensitivities and specificities reported to date for diagnosis of diseases [9]. Analysis of bodily fluids for disease diagnosis and health monitoring, from blood serum to tears, urine and saliva, represents a significantly less invasive approach than performing tissue biopsies, for example during cancer screening [10]. Such fluids are furthermore promising candidates for the application of vibrational spectroscopy, which can deliver crucial biochemical level information on patient health and indicate the presence of numerous pathologies [10,11]. Blood serum, in particular, potentially carries a wealth of important histological information whose determination could serve to improve early disease detection. Saliva has been employed for oral cancer and dysplasia diagnosis [12,13]. Urine has been investigated as liquid biopsy for cancer diagnosis [12]. In order to avoid protein/peptide species interfering with the SERS effect, an extensive fractionation should be employed prior to analyses [10,14,15,16]. In the case of saliva and urine, filtration can remove all of unwanted mucous and endothelial cells and any microbial species coexisting in the urine during collection to avoid the extra signal from other organism [5].

However, the analysis of bodily fluids is considered challenging, to a large degree due to the high dynamic concentration range of constituent protein/peptide species. While SERS, using colloidal gold or silver nanoparticles, has been used extensively to characterise blood fluids for diagnostic applications [17,18,19], it has been pointed out that the NP surfaces may have a more specific affinity for some biomolecules present in the complex millieu of the biofluid than others, and that therefore the measured SERS spectrum can be dominated by them [20]. While this may limit the sensitivity of the technique for some applications, uric acid and hypoxanthine have been identified as molecules with a particular affinity to gold and silver NP surfaces, and therefore the analysis should be particularly sensitive to any diagnostic procedure based on variations in the concentrations of these molecules [21].

Uric acid and hypoxanthine are low molecular weight constituents, present in trace amounts in body fluids [17], suggesting that they may be ideal targets for analysis the micromolar level with SERS [11]. SERS has been employed to quantify uric acid concentration in human urine [10] and both uric acid and hypoxanthine in serum and gingival crevicular fluid [11,12,13]. The studies indicate the possibility of clearly detecting uric acid and hypoxanthine in concentrations at the micromolar level with SERS [11]. To date, no research has reported the concentration of uric acid and hypoxanthine at the same time for serum, saliva and urine with SERS.

The shape of a nanoparticle is key to generating the surface enhance Raman signal [22,23]. Furthermore, control over the number of local field hotspots of gold nanoparticles can optimise the SERS efficiency [24]. In order to enhance the Raman signal, the interaction between the molecule of interest and the nanoparticles should also be considered [25,26]. The SERS effect relies on a local enhancement of the electric field due to the surface plasmon resonance (SPR) of the nanoparticles. When using spherical metal nanoparticles as SERS substrates, at least a partial aggregation is necessary to form nanoparticle clusters having the suitable plasmonic properties in order to obtain a strong SERS effect [27,28,29]. However, aggregation is not a well-controlled phenomenon and adds further uncertainty and variability to an already complex system. An alternative way to increase the local electromagnetic field associated with the SPR is to increase the local curvature of nanomaterials to produce intrinsic hotspots [23]. Nanostars contain a higher number of sharp corners and edges, and they have their own unique character, as more complex anisotropically shaped nanoparticles, and the modes oscillate at markedly different frequencies in both Au and Ag materials [23,27,29]. These modes originate from the degree and direction of polarisation of the electron cloud relative to the incident electric field to enhance Raman signals [28,29]. It has been demonstrated that such gold nanostars elicit a higher SERS effect for higher concentration of organic dye molecules [22].

The present method employs a further improvement of a previous study in which gold nanostars have been developed to enable rapid and reproducible analysis with SERS [22]. At the same time, 3kDa centrifugal filtration has been demonstrated to produce reproducible results for serum spectra analysis [5,10]. The objective of the present work is to investigate whether the same strategy could be applied to capture spectra of uric acid and hypoxanthine from human serum, saliva and urine, and to compare the analytical information attainable from HPLC. This study will therefore explore the combination of SERS using gold nanostars and fractionation using 3kDa centrifugal filtration for the analysis of the low molecular weight (LMW) fraction of three bodily fluids, blood serum, saliva and urine, from human subjects. It will demonstrate the acquisition of spectral responses, compare the concentration of urea acid and hypoxanthine against HPLC analysis.

## 2. Materials and Methods

### 2.1. Materials

AuCl_4_H_2_O, sodium citrate, hypoxanthine, hydroquinone, 10 nm gold colloid suspensions (6 × 10^12^ /mL) and hypoxanthine were purchased from Sigma Aldrich (Ireland). Sterile filtered human serum from normal mixed pool (off the clot) was purchased from TCS Biosciences (Ireland). Uric acid was purchased from FUJIFILM Wako Pure Chemical Corporation.

Five saliva and five urine samples were collected from volunteers from the Technological University, Dublin. The participants were informed that the study is scientifically relevant and not invasive, and their actions would contribute to developing a cheap, fast and simple method, which will improve population health. Verbal informed consent was provided by the participants, who agreed to contribute anonymously.

Commercially available centrifugal filtering devices, Amicon Ultra-0.5 mL (Millipore—Merck, Darmstadt, Germany), with cut-off points at 3 kDa, were employed in this study, as examples.

### 2.2. Gold Stars Synthesis and Characterisation

Gold nanostars were synthesised through the seeding growth approach, according to [30]. Briefly, 1.9 × 10^8^ mol of HAuCl_4_ were mixed with 9.3 × 10^−9^ mol of GNP seeds at room temperature (10 mL); 7.5 × 10^−9^ mol of sodium citrate, dissolved in 0.2 mL ultrapure water, were added to the mixture and stirred for 2 min in order to homogenise. Then, 3.0 × 10^−5^ mol of hydroquinone (1.5 mL) were rapidly added to the solution, under stirring. The solution was kept under stirring at room temperature for a further 20 min, after which the solution exhibits a blue colour, which is consistent with the SPR band observed.

A Perkin Elmer Lambda 900 UV/VIS/NIR Spectrometer and Zetasizer Nano ZS analyser (Malvern Instruments, Worcestershire, UK) were used to measure the absorbance, hydrodynamic particle size and zeta potentials of the nanoparticles. The samples were characterised by Electron Microscopy using a Hitachi SU6600 FESEM instrument at an acceleration voltage of 25 kV. Scanning EM images were taken using the SE detector and the Scanning Transmission EM images were taken using the TE detector. Nanoparticles were deposited on a 300-mesh grid for TEM (Ted Pella Formvar/Carbon type B) by drop casting 10 μL of the aqueous solution of nanoparticles, and allowing it to air dry. Similar nanoparticle solutions were dropped onto prewashed silicon substrates and spin coated at a speed of 1000 rpm for 20 s for SEM.

### 2.3. Fractionation of Biofluids by Centrifugation

As indicated by the manufacturer, the ultrafiltration membranes in Amicon^®^ Ultra-0.5 devices “contain trace amounts of glycerine”. The centrifugal devices were therefore washed prior to serum analysis using the protocol of [5]: The Amicon Ultra-0.5 mL filter was spun thrice with a solution of NaOH (0.1 M), followed by 3 rinses with Milli-Q water (Millipore Elix S). For both washing and rinsing, 0.5 mL of the respective liquid was added to the filters and the centrifugation was applied for 10 min at 14,000× *g* followed by a spinning with the devices upside down at 1000 g for 2 min in order to remove any residual solution contained in the filter. The procedure for 3 kDa filtering is as follows; 0.5 mL of the respective biofluid was added to the 3 kDa filter devices and the centrifugation was applied for 10 min at 14,000× *g*. A total of 15 samples of serum, saliva and urine were filtered with 3 kDa devices.

### 2.4. HPLC

Physiological levels of uric acid are in the range 20–72 μg/mL [1]. Therefore, for analysis, uric acid was disolved in concentrations of 5, 10, 20, 25 and 50 μg/mL in water. The physiological levels of hypoxanthine are in the range 50–300 ng/mL [2], and therefore the primary standard stock solutions of hypoxanthine were made by disolving the compound in water at concentrations of 50, 100, 150, 200 and 250 ng/mL.

HPLC was employed as a gold standard for determination of the concentration of the samples. The mobile phase was made up of 900 mL of HPLC grade water and 100 mL of HPLC grade Acetonitrile and was vacuum filtered using a 0.22 micron filter then sonicated for 30 min to remove trapped air bubbles. The solvent tube was placed into the mobile phase and the HPLC vials containing the different concentration of uric acid and bodily fluid filtrates were loaded into the HPLC injection compartment.

Using a C18 column (5 µm, 250 mm × 4.6 mm, Phenomenex), 10 µL of each of the filtered samples were injected by the autosampler and analysed using a photodiode array (PDA) detector. The mobile phase was water/methanol (MilliporeSigma) (50:50 *v*/*v*) with a flow rate of 1.0 mL/min. The PDA detector was operated at an excitation wavelength of 210 nm and emission wavelength of 435 nm. After the uric acid peak was eluted at 7 min, the column was eluted at a flow rate of 1 mL/min with an isocratic solvent using Methanol/water (8:2 *v*/*v*). After the hypoxanthine peak was eluted at 8.4 min, the column was eluted at a flow rate of 1 mL/min with an isocratic solvent using Methanol/water (8:2 *v*/*v*).

### 2.5. Sample Preparation for SERS Measurements

A volume of 475 μL of sample solution, either 3kDa filtered serum, urine or saliva or the different concentrations of uric acid/hypoxanthine was micropipetted into a 1.5 mL open topped 1 cm polypropylene cuvette, which was placed under the Raman microscope for spectral acquisition. Serum was diluted in a ratio of 1:4, as physiological levels of hypoxanthine of serum are general 4 times higher than those of urine and saliva [3,31,32]. A measurement of 25 μL of aqueous colloidal gold nanostars were mixed in biofluid–substrate ratios of 1:19 for a total volume of 500 μL (i.e., 25 μL:475 μL), determined to be the optimum ratio in previous studies [22].

### 2.6. Spectroscopic Measurement

Raman spectroscopy was performed using a HORIBA Jobin Yvon HR800 spectrometer with a 300 mW 785 nm diode laser as source, producing ~115 mW at the sample. The laser was focused within the liquid using a ×10 objective and the response was optimised using the water band at ~3300 cm^−1^. Spectral data were collected over the range 400–1800 cm^−1^. The accumulation conditions of Raman analysis were 10 s. The detector used was a 16-bit dynamic range Peltier cooled CCD detector [15,22,33].

### 2.7. Data Analysis

First, the peak area of HPLC were correlated with the initial concentrations to demonstrate that the concentrations of the filtered uric acid or hypoxanthine correlated with that of the solutions which were initially made up. Raman spectra obtained from the automatic mapping of different concentration of uric acid, hypoxanthine and all bodily fluids, samples were first processed using LabSpec 6 software. The Raman spectra were imported into Excel for data analysis. Linear correlations were calculated for the maximum peak intensity of SERSs spectra and the concentrations of uric acid, hypoxanthine determined by HPLC for three bodily fluids. The Wilcoxon signed-rank test was applied to the comparison of two repeated or correlated data whose measurements were at least ordinal [34].

## 3. Results

### 3.1. Characterisation of Gold Nanostars

In suspension, the nanostars show SPR bands at ~620 nm and ~700 nm (Figure 1a). Dynamic light scattering (DLS) analysis of the nanostar solutions indicates a monomodal dispersion with a hydrodynamic diameter of 45 ± 4 nm and a zeta potential of −18.5 ± 0.4 mV (Figure 1b).

TEM and SEM were employed to confirm the mophology of the nanostars. Nanostars typically have a central core and 6 vertices in a 3-dimensional arrangement. The length of the vertices is on average 10 nm and the angle at the vertex is less than 30 degrees (Figure 2).

### 3.2. HPLC and SERS of Diluted Series of Uric Acid and Hypoxanthine

The uric acid peak of HPLC was eluted at 7 min (Figure 3a). Raman spectra are shown for the series dilution of filted uric acid with a mixture ratio of 1:19 (Figure 3b), exhibiting prominent peaks at 442, 712, 802, 1000, 1085, 1206, 1343, 1436 and 1560 cm^−1^.

Figure 4a–c shows the analysis of the two sets of experimental data, along with the regression calculated from the area of the HPLC and the maximum intensity of peaks of SERs, against concentration. In each case, the data points were fitted resulting in a linear fitting curve indicated that a linear behavior can be found versus concentration in the range from 5 to 50 μg/mL. The area of the HPLC peak was seen to be linear in concentration, as shown in Figure 4a, and this linear fit can thus act as a calibration curve for the analysis of the uric acid content of the biofluids. There was a linear correlation, with R^2^ of 0.994, between the peak area of HPLC and maximum peak intensities at 1000 cm^−1^, as shown in Figure 4c. The high value of the correlation coefficient indicates a high degree of linearity of the calibration in Figure 4a.

The hypoxanthine HPLC peak was eluted at 8.4 min, as shown in Figure 5a. The SERS spectra of the series dilution of hypoxanthine with nanostars are shown in Figure 5b. The prominent peaks appear at 560, 640, 803, 1206, 1290 and 1620 cm^−1^.

Figure 6 shows the analysis of the two sets of experimental data, along with the regression calculated from the area of HPLC, and the maximum intensity of SERS peaks against concentration. The high value of the correlation coefficient indicates a good linearity of the calibration of HPLC vs. concentration in Figure 6a, and this linear fit can thus act as a calibration curve for the analysis of the hypoxanthine content of the biofluids. For the SERS peaks, a linear behavior can be found when the peak intensities were plotted versus concentration over the range from 50 to 250 ng/mL (Figure 6b). The peak intensities at 640 cm^−1^ show the highest R^2^. There was a linear correlation between peak area of HPLC and SERS peaks intensities, as shown in Figure 6c for the case of 640 cm^−1^, with an R^2^ value of 0.959.

### 3.3. SERS of 3 kDa Filtrate of Bodily Fluids

The SERS spectra of the 3 kDa filtrate of serum, saliva and urine with a nanostars mixture ratio of 1:19 are shown in Figure 7. In each case, the SERS spectra of five different samples showed a high degree of reproducibility, as shown in Figure 7.

### 3.4. Overlapping Spectral Peaks of SERS of Uric Acid, Hypoxanthine and Body Fluids

Figure 8 compares the average spectra observed for the three bodily fluids and that of the solutions of the pure compounds, uric acid and hypoxanthine. The vertical blue solid lines indicate peaks of uric acid which are evident in the bodily fluids, and the vertical dashed lines, those of hypoxanthine.

There is a significant overlap of the spectral bands of the SERS spectrum of uric acid at 442, 712, 802, 1000, 1086, 1206, 1343, 1436, 1560 cm^−1^ with those of the three body fluids at spectra peaks. The features at 560, 640, 802, 1206, 1206 and 1620 cm^−1^ from hypoxanthine were also evident in the spectra of serum, saliva and urine. The assignment of SERS spectral peaks of the different samples was listed in Table 1.

The SERS maximum values at 1000 cm^−1^ were analysed for the three body fluids and different concentrations of uric acid. The maximum values were employed to calculate the uric acid concentration from the standard curve in Figure 4, as listed in the left panel of Table 2. The concentrations of uric acid obtained by SERS showed good agreement with the uric acid concentration as measured with HPLC analysis of the respective body fluid, shown in the right hand panel of Table 2. For all biofluids, the concentration range was from 8 to 23 μg/mL. The Wilcoxon signed-rank test of the experimental intercept showed that this did not differ significantly from the two experiments. The *p*-value is 0.4354.

In the case of hypoxanthine, the 640 cm^−1^ peak was selected as optimal for determination of hypoxanthine concentration, as it exhibited the highest linear correlation with the peak area of HPLC in Figure 5. The spectral peak maxima were employed to calculate the hypoxanthine content for each of the bodily fluids, as shown in Table 3. The concentration range was from 31 to 170 ng/mL. The Wilcoxon signed-rank test of the experimental intercept showed that this did not differ significantly from the two experiments. The *p* value is 0.2801.

## 4. Discussion

The current study uses gold nanostars with a hydrodynamic diameter of 45 ± 4 nm and a zeta potential of −18.5 ± 0.4 mV in a monomodal dispersion (Figure 1). The gold nanostars typically have a central core and six vertices in a three-dimensional arrangement (Figure 2). The gold nanostars enable determination of the concentration of uric acid and hypoxanthine in the range 5–50 μg/mL and 50–250 ng/mL, respectively (Figure 3 and Figure 5). The peak area of HPLC and maximum peak intensity of SERS are strongly correlated. The maximum peak intensities of 1000 cm^−1^ show linear correlation with the peak area of HPLC of uric acid. R^2^ was 0.975 (Figure 4). The maximum peak intensities of 640 cm^−1^ show linear correlation with the peak area of HPLC of hypoxanthine. R^2^ was 0.959 (Figure 6). Our current results suggest good reproducibility for the 3 kDa centrifugally filtered biofluid samples, measured for 10 s in 19:1 body fluid-substrate mixtures with gold nanostar solution using a 785 nm laser source.

Bodily fluids are promising candidates for the application of vibrational spectroscopy, which can deliver crucial biochemical level information on patient health and indicate the presence of numerous pathologies [9,10]. Blood serum in particular potentially carries a wealth of important histological information whose determination could serve to improve early disease detection. Saliva has been employed for oral cancer and dysplasia diagnostic [9]. Urea is the main molecular constituent and has similarly been investigated as liquid biopsy for cancer diagnostic [10]. Notably, despite significant differences in the biochemical composition, the dominant features in the SERS spectra of three bodily fluids are remarkably similar (Figure 7). This observation is consistent with that of Inacu and coauthors that both serum and urine show striking similarities in their SERS spectra, providing similar metabolic information [10]. The SERS spectra of 3 kDa filtered bodily fluids have been compared to those of uric acid and hypoxanthine in Figure 8. Uric acid has peaks at 442, 712, 802, 1000, 1086, 1206, 1343, 1436 and 1560 cm^−1^ (Figure 5), all of which can be detected in the spectra of all three biofluids, urea, saliva and serum (Figure 8 and Table 1). Similarly, features of the SERS spectrum of hypoxanthine at 560, 640, 802, 1206 and 1560 cm^−1^ can be identified in the spectra of all three bodily fluids (Figure 8 and Table 1). Based on a standard calibration curve of concentration-dependent SERS spectra, the strongest peaks, at 1000 cm^−1^ and 640 cm^−1^, for uric acid and hypoxanthine, respectively, can be employed to determine the concentrations of the compounds in each bodily fluid. An excellent agreement is observed for the concentration determined by SERS and HPLC analyses for both constituent compounds, as shown in Table 2 and Table 3. Minimum concentrations of 8 μg/mL were determined for uric acid in serum, and 33 ng/mL of hypoxanthine in serum.

Our result is consistent with that of Premasiri and coauthors, who observed that the SERS spectra can be employed to detect trace levels of hypoxanthine in whole blood and plasma [3]. However, the storage period influences the vibrational features of hypoxanthine [38]. The complexity of the serum proteome presents challenges for efficient sample preparation and adequate sensitivity for analysis [39]. For the liquid SERS analysis of purine metabolites from serum, proteins have to be removed since the relatively high concentration of serum proteins prevents the adsorption of metabolites [3,38]. Note, the use of centrifugal filtration to fractionate the biofluids is largely based on previous works [40,41]. Recently, Esposito and co-workers have proposed an alternative technique to spin gingival crevicular fluid on Periopaper^®^ filter paper to undergo SERs analysis [42].

The choice of nanostars is based on prior experience of some of the authors [22]. When using gold nanomaterials, one drawback might be the high photothermal conversion efficiency provided under resonance excitation, which can be destructive for biological samples. Thus, the rational choice of the shape and composition, and tuning of the excitation spectral window for minimal heating and photodegradation are key elements in using nanostructured probes in biosensing [39]. The intrinsic hotspots of gold nano stars gave quite different spectral peaks compared to assign bands in SERS spectra of chemicals, serum and plasma in literature [3,12,19]. The current study shows that the limits of detection (LOD) for uric acid are consistently higher than those for hypoxanthine, across all biofluids, using the same SERS approach. Westley et al. determined the (LOD) for uric acid to be 5 μM in urine [18]. However, Premasiri and coauthors reported that the LOD hypoxanthine in blood is 1 nM [38]. Uric acid and hypoxanthine have different solubilities in bodily fluids [19,35] and the significant differences in LOD may be due to different affinities of the two molecules to bind to the gold surface [39]. Nevertheless, the method of peak fitting provides a high degree of accuracy within the range of concentrations of physiological relevance [1,35].

The disease burden of hyperuricemia is increasing, especially in high-income countries and economically developing world with a Western lifestyle [2]. An in-crease of the serum uric acid and hypoxanthine concentration in patients with myocardial infarction indicates a significant metabolic involvement of xanthine oxidoreductase in this disease and therefore a possible role in the development of tissue damage in the postischaemic phase due to oxygen radicals generated by the oxidase activity of this enzyme [40,41]. Increased uric acid and hypoxanthine concentration is associated with cancer, HIV, gout, and hypertension. In contrast, low levels are associated with Alzheimer disease, progression of multiple sclerosis, and mild cognitive impairment [3]. Uric acid or hypoxanthine sensing research based on various techniques such as ion sensitive field effect transistors, optical fibre-based SPR, and electrochemical system with different nano materials has been explored [42,43,44,45]. The ability to accurately measure levels of uric acid and hypoxanthine is important for purine metabolism disorders and cancer diagnostics [46,47,48,49], and, increasingly, simultaneous metabolic profiling of multiple biofluids has been recognised as a key strategy in biomarker discovery research [9] and ultimately potentially beneficial for improved treatment and prognosis [39].

Furthermore, there is excellent agreement between the results of the two analytical approaches, SERS and HPLC, for all samples analysed, as indicated in Table 3. The calibration curves of Figure 4 and Figure 6 are based on maximum peak intensities, and pure solutions of the respective compounds. Bonifacio and coworkers have demonstrated that uric acid and hypoxanthine are major contributors to the SERS spectrum of bodily fluids [20,21]. Two research groups have reported independently that there are strong linear correlations between concentration for uric acid or hypoxanthine with HPLC in bodily fluids [18,38]. The current work further confirms that the concentrations of the two species can be accurately determined, based on the calibration curves which correlate the maximum peak intensity with the concentration, as determined by HPLC. The methodology could be further explored and potentially improved by a multivariate regression correlation using the full spectrum, and or an examination of multicomponent solutions, to explore competitive binding effects [50]. Nevertheless, it is suggested that SERS provides a potentially valuable substitute for HPLC-based methods for the detection, identification, and quantification of urine acid and hypoxanthine in serum, saliva and urine for a variety of medical applications.

Next-generation analytical methods may rely on SERS of filtrated samples to deliver patient’s health information crucial for clinical decision-making and indicate the presence of numerous pathologies sufficiently early to improve patient prognosis [51,52,53,54]. A SERS method to increase the sensitivity to small and low-concentrated molecules in biofluids may help screening of a variety of diseases and organ impairments/failures [55,56,57,58,59,60]. Currently, protein sample are run through an extended liquid chromatography (LC) gradient, and the eluent is analysed by electrospray mass spectrometry (MS/MS). The LC-MS/MS for analysing proteome can cost around €500,000. It also requires expensive running reagents, a specific column and a trained practitioner to operate equipment. A single sample analysis costs over €200–2000. In contrast to the LC-MS/MS, SERS analysis shows significant promise as a rapid (10 s) low-cost (~€1 per sample) technique. The substitution of a SERS based methodology for detection of hypoxanthine and uric acid offers advantages relative to HPLC approaches. It requires minimal sample handling as compared to the HPLC approach. No deproteinisation step is necessary, no additional chemical treatment steps are needed for detection, and no instrument calibration procedures, as required for HPLC-based identification techniques, need to be performed [38]. Both chromatographic and SERS based techniques are inherently multiplexing detection schemes. The three body fluids have common spectra in urea acid and hypoxanthine can help disease diagnostic [61,62]. In future, analysis of other biofluids out of blood or biopsy can be employed to detect biomarkers for the diagnosis of bodily pathologies. It is clear that current biochemical/biophysical analysis methods provide light for rapid and low-cost especially in economically underdeveloped areas [63,64].

In terms of future perspectives, conjugated nanoparticles can target specific antigens for disease diagnostics. Colorimetric assays and imaging analysis have been developed [65]. Research using label-free SERS analysis should overcome the enormous discriminative and analytical potential to close to an actual clinical application. For example, increase quality of samples by filtration to combine Raman system to increasingly miniaturise and make the system more robust and stable; and provide quick and user-friendly data analysis, so that clinicians can quickly understand the results found [66].

## 5. Conclusions

The SERS method is an alternative to add sensitivity to current Raman spectroscopy methods aiming for high-throughput screening, as little expertise of specialised professionals will be required once the sample preparation protocol is established and sample analysis is automated by machine learning algorithms. It is first time that correlation of results between SERS and HPLC on serum, saliva and urine have been reported. The results illustrate the first attempt to establish such a protocol, which has been optimised to analyse body fluids after 3 kDa centrifugal filtration, based on a 10 s measurement of 19:1 body fluid-substrate addition of gold nanostar solution. Characteristic peaks of uric acid and hypoxanthine are clearly identifiable in the spectra of all three biofluids, urea, saliva and serum and thus SERS can be used to quantify their content. The analysis has a high degree of accuracy over physiologically relevant ranges The study demonstrates that SERS is a very attractive alternative to HPLC-based methods for monitoring uric acid and hypoxanthine in biofluids, for the diagnosis and study of a number of diseases or conditions such as hyperuricemia.

## Figures and Tables

**Figure 1 nanomaterials-13-01216-f001:**
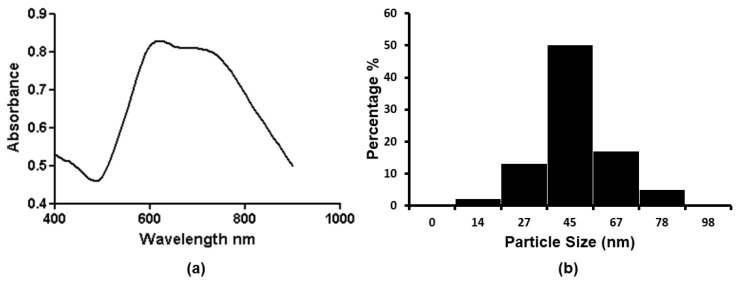
UV-vis absorbance spectrum (**a**) and particle size distribution (**b**) of AuNS solution.

**Figure 2 nanomaterials-13-01216-f002:**
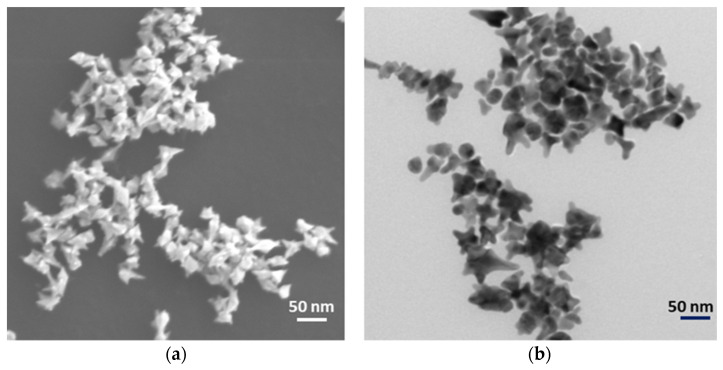
Nanostars under (**a**) Scanning and (**b**) Transmission Electron microscopy.

**Figure 3 nanomaterials-13-01216-f003:**
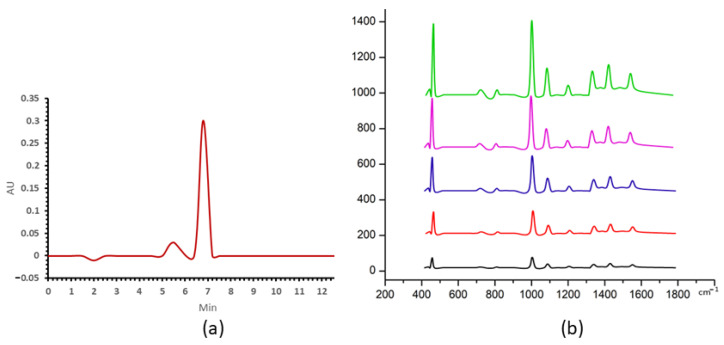
(**a**) The uric acid HPLC peak at concentration of 50 μg/mL, (**b**) SERS spectra of uric acid for concentrations at 5, 10, 20, 25, 50 μg/mL in black, red, blue, purple and green, respectively.

**Figure 4 nanomaterials-13-01216-f004:**
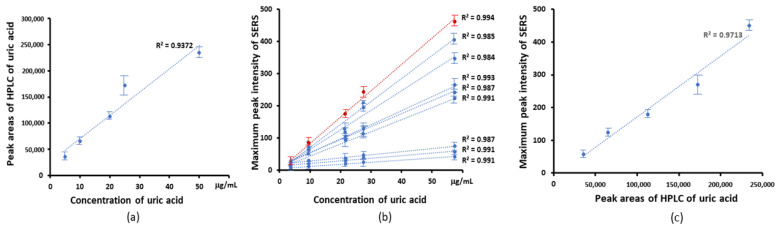
Linear correlation of HPLC and SERS with the concentration of uric acid conentration. (**a**) The linear correlation of HPLC peak areas with concentration of uric acid. (**b**) The maximum peak intensity of SERS vs. the concentration of uric acid. The linear correlation from top to bottom is maximum peak at 1000, 442, 1436, 1085, 1343, 1560, 802, 1206, 712 cm^−1^ to concentration of uric acid. The peak at −1000 cm^−1^ (red line) showed a linear concentration with. R^2^ of 0.994 (**c**) The linear correlation of the maximum peak intensity of SERS and peak areas of HPLC of uric acid. R^2^ was 0.971.

**Figure 5 nanomaterials-13-01216-f005:**
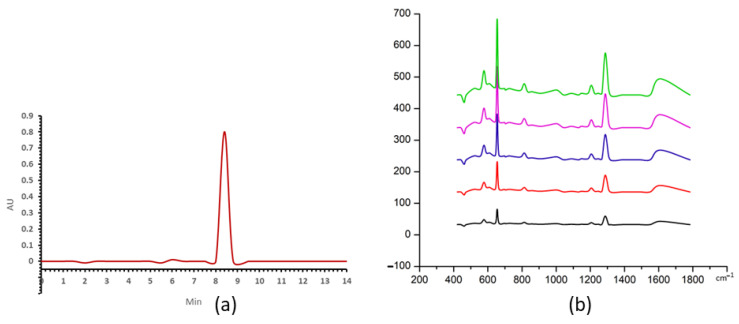
(**a**) The hypoxanthine HPLC peak of 250 ng/mL, (**b**) SERS spectra of hypoxanthine for concentrations at 50, 100, 150, 200 and 250 ng/mL in black, red, blue, purple and green, respectively.

**Figure 6 nanomaterials-13-01216-f006:**
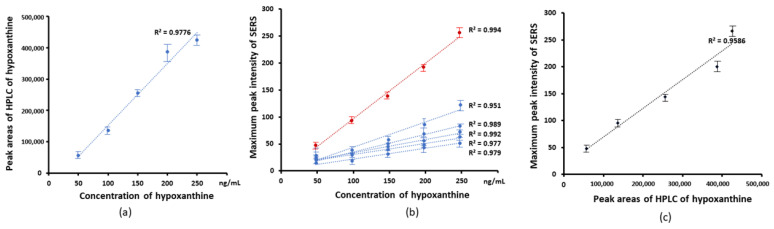
(**a**) The linear correlation of HPLC peak areas versus concentration of hypoxanthine (**b**) Correlation of SERS peak intensities versus concentration of hypoxanthine, from top to bottom. The maximum peak at 640, 1290, 560, 1620, 1206, 802 cm^−1^. The peak at ~640 cm^−1^ (red line) showed an R^2^ of 0.994. (**c**) The correlation of 640 cm^−1^ maximum peak intensity vs. peak areas of HPLC for hypoxanthine. R^2^ is 0.959.

**Figure 7 nanomaterials-13-01216-f007:**
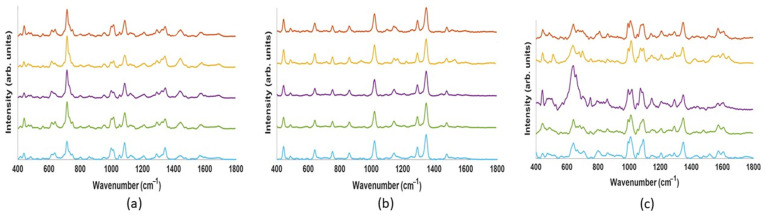
Repeatability of SERS spectra of 3 kDa filtrates of human serum, saliva and urine samples with added nanostars (nanostar: sample = 1:19 in volume ratio). (**a**) From five independent experiments of serum. (**b**) Spectra from five different saliva samples, that from each donor depicted in a different colour. (**c**) Spectra from five different urine samples, that from each donor depicted in a different colour. For all spectra, source wavelength was 785 nm, laser power 125 mW, and acquisition time was 10 s per spectrum.

**Figure 8 nanomaterials-13-01216-f008:**
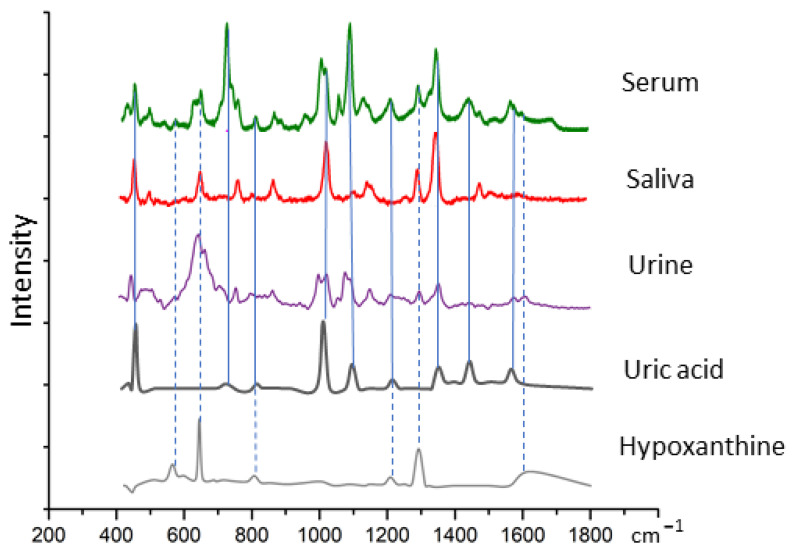
Overlap of spectra of 3 kDa filtrates of human serum, saliva and urine with uric acid and hypoxanthine. The intensity of peaks and highest overlap peak in body fluids and uric acid and hypoxanthine. The overlap peaks with uric acid in solid blue line. The overlap peaks with hypoxanthine in vertical dash blue line. The assignment of peaks is provided in Table 1.

**Table 1 nanomaterials-13-01216-t001:** Assignment of SERS of uric acid, hypoxanthine and 3 kDa filtration of bodily fluids.

SERS cm^−1^			Assigment of Spectral Features		
Serum	Saliva	Urine		Uric Acid	Hypoxanthine
1620	1620	1620	ν ring vibration [35]	−	+
1560	1560	1560	vibration C−N [10]	+	−
1436	1436	1436	(C_2_ H_2_) [10]	+	−
1343	1343	1343	ν ring vibration [3,36]	+	−
1290	1290	1290	(C_2_ H_2_) [36]	−	+
1206	1206	1206	C = O vibration [37]	+	+
1086	1086	1086	(NH_3_), (CH_2_) [10]	+	−
1000	1000	1000	symmetric C-N stretching [10]	+	−
860	860	860	side chain vibrations [37]	−	+
802	802	802	(CH_2_) [38]	+	+
712	712	712	N−H bending [36]	+	−
640	640	640	skeletal ring deformation [35]	+	−
560	560	560	ring vibration [37]	−	+
442	442	442	(NH_2_) [10]	+	−

**Table 2 nanomaterials-13-01216-t002:** SERS and HPLC methods for uric acid concentration detection (μg/mL).

	SERS			HPLC	
Serum	Saliva	Urine	Serum	Saliva	Urine
8	23	18	7	21	17
10	21	10	11	20	12
13	15	20	11	17	23
9	12	11	8	13	10
17	22	22	19	23	20

**Table 3 nanomaterials-13-01216-t003:** SERS and HPLC methods for hypoxanthine concentration detection (ng/mL).

	SERS			HPLC	
Serum	Saliva	Urine	Serum	Saliva	Urine
43	53	67	47	51	61
37	57	78	41	50	72
33	47	170	31	47	153
55	47	70	58	43	70
53	52	87	59	53	85

## Data Availability

The data presented in this study are available online within this article.

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
