# Peer review of "Surface-Enhanced Raman Analysis of Uric Acid and Hypoxanthine Analysis in Fractionated Bodily Fluids"

_nanomaterials, 2023, doi:10.3390/nano13071216_

Round 1

Reviewer 1 Report

The authors have used the gold nanostar for label-free metabolite detection in 3 kinds of body fluids and demonstrated the high correlation between SERS and HPLC results. But several important issues have not been fully discussed, which have to be supported with additional data and explanations before being published.

1.     The author highlighted that there is no article reporting the quantification of uric acid and hypoxanthine at the same time in 3 kinds of body fluids, but the author didn’t provide any discussion or data upon the benefits that could be gained from this point.

2.     Why does serum need dilution before measurements while others don’t? Why are the samples mixed with the Au Nanostars in the ratio of 1:19?

3.     Why were the Au Nanostar characterized to be 18.15 mV? Au Nanostars were reduced by citrate and predicted to negatively charged.

4.     The caption and the legend in Figure 3 don’t match.

5.     Statistical data (e.g., error bars) should be provided for of the calibration curves in Figure 4 since reproducibility and accuracy are among the major concerns of quantification.

6.     For body fluids which consist of abundant metabolites, each peak can be attributed by a wide range of molecules. Simply attributing one peak to one single metabolite is not appropriate. Moreover, complicated co-adsorption of different metabolites onto the nanoparticles will also greatly vary the response from that gained from pure samples at certain concentration. Using the calibration curve obtained from single-analyte samples to quantify the molecules in a mixed sample is also improper. The high correlation between HPLC and SERS required more discussion.

7.     There are several spelling mistakes. Please recheck the manuscript.

8.     A very closed review article (Human metabolite detection by surface-enhanced Raman spectroscopy, Materials Today Bio, 2022, 13, 100205.) should be cited.

Author Response

Dear Reviewer,

Thank you very much for taking your time to review the manuscript. The comments are very helpful. We have taken this chance to revise the manuscript. The responds are in red colour in revised manuscript.

Looking forwards to receive your reply

Furong

Reviewer 2 Report

The authors claim to report the simultaneous determination of the concentrations of uric acid and hypoxanthine for serum, saliva and urine, by using prior filtration and SERS.

They are supporting their results based on a standard calibration curve of concentration dependent SERS spectra, the strongest peaks, at 1000 cm-1 and 640 cm-1, for uric acid and hypoxanthine respectively, that can be employed to determine the concentrations of the compounds in each bodily fluid.

One important aspect that makes this study different form other SERS approaches reported is that no deproteinisation step is necessary, no additional chemical  treatment steps detection, and no instrument calibration procedures, as in the case of HPLC-based identification techniques, have to be performed.

I agree with the authors that this might be one attempt more towards SERS standardization and that more universal next-generation analytical methods relying on SERS of filtrated samples will be developed in order to deliver patient’s health information crucial for clinical decision-making.

The technical questions I would ask the authors are:

Why use prefer to use filtration instead of protein precipitation as in previous reports cited by the authors? The time required is about the same and precipitation might be less expensive than some special filters as cost per analysis. Is there another advantage of this sample preparation approach?

Why use gold nanostars when their EF in this particular study does not exceed the silver-based nanoparticles reports? (cited by the authors as Iancu et al., Stefancu et al., Moisoiu et al.) These previous studies were the fundamental principle for more studies that have to deal with cancer early diagnosis, so there is a lot of potential in their procedure based of simple, low cost silver NPs. Why then using gold? (Also in terms of cost /analysis that will increase it..)

Another technical aspect: why is the LOD for uric acid so different to the hypoxanthine one for the same SERS approach? Wouldn’t this limit the further envisioned clinical applications in many ways?

 A final suggestion: instead of citing the ref 59, Chaudhary et al Clinical Spectroscopy 2022, 4, 100022, the authors can consider to cite more relevant, recent work: doi 10.1016/j.trac.2020.116122; doi 10.1016/j.colsurfb.2021.112064; doi 10.1039/D1NR00708D; doi 10.1016/j.bios.2022.114843 

After giving a short point to point answer list for these questions, I will be able to recommend the paper for publishing.

Author Response

Dear Reviewer,
Thank you very much for taking your time to review the manuscript. The comments are very helpful. We have taken this chance to revise the manuscript. The responds are in green colour at below and in revised manuscript.

best regards

Furong 

Round 2

Reviewer 1 Report

No further comments.